# The Advantage of Immunohistochemical Staining for Evaluating Lymphovascular Invasion Is Limited for Patients with Esophageal Squamous Cell Carcinoma Invading the Muscularis Mucosa

**DOI:** 10.3390/jcm11236969

**Published:** 2022-11-25

**Authors:** Akira Dobashi, Daisuke Aizawa, Yuko Hara, Hiroto Furuhashi, Hiroaki Matsui, Toshiki Futakuchi, Shingo Ono, Hirobumi Toyoizumi, Fateh Bazerbachi, Takashi Yamauchi, Machi Suka, Kazuki Sumiyama

**Affiliations:** 1Department of Endoscopy, The Jikei University School of Medicine, 3-25-8 Nishi-Shimbashi, Minato-ku, Tokyo 105-8461, Japan; 2Department of Pathology, The Jikei University School of Medicine, 3-25-8 Nishi-Shimbashi, Minato-ku, Tokyo 105-8461, Japan; 3Centra Care, Interventional Endoscopy Program, St. Cloud Hospital, 1900 Centracare Cir Ste 2400, St. Cloud, MN 56303, USA; 4Department of Public Health and Environmental Medicine, The Jikei University School of Medicine, 3-25-8 Nishi-Shimbashi, Minato-ku, Tokyo 105-8461, Japan

**Keywords:** esophageal squamous cell carcinoma, endoscopic resection, long-term, lymphovascular invasion, immunohistochemical staining

## Abstract

The cumulative metastasis rate of esophageal squamous cell carcinoma (ESCC) pathologically invading the muscularis mucosae (pT1a-MM), based on lymphovascular invasion (LVI) evaluated by immunohistochemical (IHC) staining is unknown. This retrospective study included patients with endoscopically resected pT1a-MM ESCC. The primary endpoint was the metastasis rate of pT1a-MM based on LVI, evaluated using IHC and additional prophylactic therapy. The secondary endpoint was the identification of independent factors for metastasis based on lesion characteristics. The prognosis was also analyzed considering the impact of head and neck cancer. A total of 104 patients were analyzed, with a median follow-up of 74 months. The positive rate for LVI was 43.3% (45/104). In 33 patients, IHC was not performed at the time of clinical evaluation, 8 of whom exhibited LVI. However, these patients did not exhibit metastasis. The metastasis rates of patients without LVI, those with LVI and additional therapy, and those with LVI without additional therapy were 5.1%, 20.8%, and 0%, respectively. Lesion size ≥ 25 mm was the only independent factor for metastasis in multivariate analysis. The advantage of IHC for determining additional prophylactic therapy is limited for patients with pT1a-MM ESCC.

## 1. Introduction

Endoscopic resection (ER) is a minimally invasive treatment for superficial neoplasia of the gastrointestinal tract [1] given the lower likelihood of lymph node metastasis (LNM) [2,3]. This is particularly relevant in patients with esophageal squamous cell carcinoma (ESCC). Initially, endoscopic mucosal resection (EMR) was applied to the treatment of ESCC. Thereafter, endoscopic submucosal dissection (ESD) was developed, and ESCC is treated by ESD technique currently. The advantage of ESD is a higher en bloc resection rate compared to EMR because ESD is done by a needle knife and an endoscopist can determine the tumor margin during mucosal incision. On the other hand, the size of specimen that can be resected by EMR is limited and a piecemeal resection may be needed for a larger lesion. The en bloc resection enables precise pathological diagnosis. Once the tumor involves the muscularis mucosae (MM), it tends to spread lymphatically or hematogenously [4,5,6,7,8,9,10,11,12,13], and the risk increases proportionately with the depth of invasion. Therefore, therapy-naïve ESCC patients can be divided into three categories based on the likelihood of metastatic disease: ESCC limited to the epithelium (EP) or lamina propria (LPM), MM/submucosal tumor invasion ≤200 µm (SM1), or submucosal tumor invasion > 200 µm (SM2) [8]. Clinically evaluated EP/LPM (cT1a-EP, LPM) is an absolute indication for ER given the rare risk of metastatic disease. When pathology diagnoses the lesion as invading the EP and LPM (pT1a-EP, LPM), no additional treatment is recommended [8]. Moreover, the metastasis rate of pathologically MM-invading tumors (pT1a-MM) is also low and reported between 0–26.7%. Thus, most patients whose pathology is diagnosed as pT1a-MM ESCC without lymphovascular invasion (LVI) do not receive additional prophylactic therapy, even though the Japan Gastroenterological Endoscopy Society ESD/EMR guidelines for patients with pT1-MM ESCC without LVI do not make specific evidence-based recommendations regarding the need for additional prophylactic therapy, given that previous reports may have underestimated the metastatic risk due to low power or limited follow-up time [8].

However, precisely distinguishing pT1a-MM from pT1b-SM1 using endoscopy before ER remains challenging. Thus, cT1a-MM/cT1b-SM1 is another indication for ER and cT1a-MM and cT1b-SM1 disease are grouped together [4,5], despite the difference in the rate of metastasis between these two subgroups. However, the implementation of advanced ER techniques allows accurate assessment of the depth of involvement, and better prognostication for these patients should be considered based on the invasion depth between pT1a-MM and pT1b-SM1 [9].

LVI is one of the most important surrogate markers for predicting ESCC metastasis [7,9,10,11,12,13]. Conventional hematoxylin and eosin staining (HE)-based pathological evaluation of LVI has been used; however, it has several limitations in terms of sensitivity and reproducibility. Therefore, better techniques, such as Elastica–Van Gieson staining (EVG) and immunohistochemical (IHC) staining with anti-D2-40, CD31, or CD34 antibodies, have been adopted and reported to provide a better evaluation of LVI. When examined in surgical specimens, IHC improved the prediction of ESCC metastasis [14]. However, most studies that used endoscopically resected ESCC and showed the relationship between metastasis and LVI implemented HE staining only [4,5,7,9,10,11,13], and the effects of IHC on determining metastasis and additional prophylactic therapy are unknown.

Given the concept of field cancerization, SCC in the esophagus, head, or neck may occasionally develop synchronously or metachronously [15]. Therefore, patients with head and neck cancer (HNC) are recommended to undergo endoscopic screening and surveillance for ESCC and vice versa [16]. It was reported that 8.4% of patients who received endoscopy for staging ESCC also had HNC; therefore, the presence of HNC should be considered [17], as the presence of HNC worsens the prognosis of ESCC patients. Therefore, we aimed to reveal the effect of IHC-detected LVI for the prediction of metastasis and prognosis of endoscopically resected pT1a-MM ESCC.

## 2. Materials and Methods

### 2.1. Study Design

All patients with endoscopically resected ESCC at the Jikei University School of Medicine between January 2005 and December 2020 were prospectively captured in our registered database and analyzed retrospectively in September 2021. The study was approved by the Clinical Ethics Committee of the Jikei University School of Medicine (No. 27-222(8107)).

### 2.2. Study Group

We included patients with pT1-MM ESCC. We excluded patients with (1) ESCC treated with any modality before ER, (2) ER specimens showing vertically positive margins, (3) treatment of more than two pT1a-MM ESCCs, and (4) a history of other neoplasms treated with any form of chemotherapy or radiotherapy before ER. We excluded patients who had a history of HNC but included synchronous HNC which was treated after ER for ESCC.

### 2.3. ER

Before ER, the invasion depth was estimated endoscopically using white-light and narrow-band imaging [16,18]. Computed tomography (CT) was used to estimate lymph node involvement or distant metastasis. Endoscopic ultrasound was optionally performed when submucosal involvement was suspected. The protocol for staging assessment has been previously reported [19]. Between 2005 and 2007, cap-assisted endoscopic mucosal resection (C-EMR) was the preferred procedure; however, after 2007, endoscopic submucosal dissection (ESD) became the method of choice [20,21].

### 2.4. Histological Evaluation

Resected specimens were fixed in 10% buffered formalin and manually cut into 2 -mm slices. HE staining was performed on 4-μm slices. Histological diagnosis was established according to the Japanese classification of esophageal cancer [22]. The depth of tumor invasion was classified as EP, LPM, MM, SM1, or SM2. Complete resection (R0 resection) was defined as the absence of tumor recognition at any specimen margin. A tumor recognized in any specimen margin was defined as an incomplete resection (R1). Resection was evaluated as non-assessable (RX) when it was not possible to evaluate clear margins secondary to a diathermy effect. When ESCC extended beyond the MM, IHC staining, including anti-D2-40 antibody immunostaining and EVG staining, was performed to evaluate LVI [14]. The infiltrative growth pattern (INF) at the tumor margin was also evaluated according to the guidelines [22]. INF of the tumor was classified into three categories (a, b, or c) according to the predominant pattern observed at tumor margins. Expansive, intermediate, and infiltrative types were categorized as INFa, INFb, and INFc, respectively.

### 2.5. Additional Prophylactic Therapy and Follow-Up

Additional prophylactic therapy was recommended for all pT1a-MM patients. Esophagectomy with lymph node dissection was the first option, and radiotherapy, with or without chemotherapy, was optional. The chemoradiotherapy consisted of 60 Gy irradiation combined with chemotherapy with standard-dose 5-fluorouracil (700 mg/m^2^ (Day 1–5)) and cisplatin (70 mg/m^2^ (Day 1)). The radiotherapy consisted of 60 Gy irradiation. The dose was reduced depending on the patient’s condition. Patients were advised that the rate of metastasis was low when LVI was negative to help them decide on additional therapy. Follow-up endoscopy with Lugol chromoendoscopy was performed every 3–12 months. CT and tumor marker serology were also obtained every 3–12 months after ER, regardless of LVI or additional therapy.

All patient records were reviewed in September 2021 to verify the outcomes (prognosis). For patients not present for a six-month, in-person visit or for whom follow-up was less than five years, data were obtained through telephone interviews or by obtaining records from the referring institution.

### 2.6. Outcomes

The primary outcome was the rate of LNM or distant metastasis of pT1a-MM based on LVI, which was evaluated using IHC staining with D2-40 for lymphoid invasion and EVG for vessel invasion. The secondary outcomes were independent factors for metastasis and survival, the five-year overall survival and cause-specific survival rates based on LVI, additional therapy, and HNC status.

### 2.7. Statistical Analysis

Statistical analyses were performed using SAS (version 9.4, SAS Institute, Cary, NC, USA). Independent continuous variables were compared using the Mann–Whitney U test, whereas categorical variables were compared using the χ^2^ (chi-squared) test or Fisher’s exact test. Logistic regression analysis was used for univariate and multivariate analyses to identify factors associated with metastasis.

For prognostic analyses, patients were divided into four groups to isolate the effect of HNC. Three groups included patients without HNC, and the fourth group consisted of patients with HNC. The groups were defined as follows: patients without LVI (Group 1), patients with LVI and additional prophylactic therapy (Group 2), patients with LVI without additional prophylactic therapy (Group 3), and patients with HNC (Group 4). Patient survival was calculated using Kaplan–Meier survival analysis and compared using the log-rank test. The last follow-up was defined as either the date of death or the date of last contact obtained from medical records (in-person visit) or a telephone interview. The Cox proportional hazard model was used to evaluate significant predictors of disease-free survival and overall survival. Statistical significance was defined as a two-tailed *p*-value < 0.05.

## 3. Results

### 3.1. Patient Demographics and Short-Term Outcomes of ER

We analyzed 104 lesions in 104 patients (one lesion per patient) who met the inclusion criteria. The baseline characteristics are shown in Table 1. The mean (±SD) age was 64.8 ± 8.6 years, and 94 patients (90.4%) were males. A history of HNC was present in 28.8% of patients. The median follow-up period was 74 months (range: 6–184). Additional therapy was administered to 24 patients: 5 underwent esophagectomy and lymph node dissection, 10 received radiotherapy, and 9 received chemoradiotherapy. At the end of follow-up, 21.2% of patients had died. The mean (±SD) lesion size was 29.7 ± 16.4 mm, and 19 lesions were treated by EMR, while 85 lesions were treated by ESD. The en bloc resection rate of ESD was significantly higher than that of EMR (68.4% [13/19] vs. 100% [85/85], *p* < 0.01). The R0 resection rate of ESD was also significantly higher than that of EMR (73.7% [14/19] vs. 95.3% [81/85], *p* < 0.01). However, no local recurrence was observed during the follow-up period.

### 3.2. Lymph Node or Distant Metastasis Based on LVI Evaluated by IHC

HE staining, but not IHC, was performed in 33 patients at the time of clinical evaluation. Therefore, IHC was requested for these cases. As a result, eight cases previously negative for LVI by HE staining showed LVI by IHC (Table 2). The detection of LVI increased from 15.1% (using HE: 5/33) to 34.9% (adding IHC: 13/33). The overall detection of LVI by IHC was 43.3% (45/104). The overall detection of lymph nodes or distant metastases during follow-up was 8/104 (7.7%). This rate was compared between LVI-negative patients (5.1%; 3/59) and LVI-positive patients (11.1%; 5/45) (*p* = 0.28) (Table 3). Before IHC was requested for these 33 patients, there were 49 patients in Group 1. The detection of LVI in these eight patients subsequently moved them to Group 3. Moreover, none of the three patients with metastases in Group 1 were shown to have LVI by IHC. As a result, the metastasis rate increased from 6.1% (3/49) to 7.3% (3/41) in Group 1.

### 3.3. Clinical Course after ER

Groups 1, 2, 3, and 4 comprised 41, 18, 15, and 30 patients, respectively. The clinical course of these groups is shown in Figure 1, and the demographics of each group are shown in Table 1.

At the end of follow-up, the respective mortality rates in Groups 1, 2, 3, and 4 were 12.2%, 5.6%, 20.0%, and 43.3%, and the respective recurrence (LNM or distant) rates were 7.3%, 22.2%, 0%, and 3.3%. Among the patients in Group 1 as well as the 7.3% (3/41) of patients who experienced LNM or distant metastasis, we added additional pathology stains to confirm the lack of LVI, and none had LVI with the additional IHC staining.

In Group 2, the modalities of additional therapy were esophagectomy (*n* = 4), radiotherapy (*n* = 8), and chemoradiotherapy (*n* = 6). Three of four patients who underwent esophagectomy had LNM and received additional chemotherapy after surgery. One patient who received radiotherapy (60 Gy) had ESCC liver metastases and died. In Group 3, none of the patients had lymph nodes or distant metastasis during follow-up by CT. In Group 4, 21 patients received chemotherapy or radiotherapy for HNC after ER. One patient who underwent esophagectomy for additional therapy had LNM. The main cause of mortality in Group 4 was HNC (61.5%, 8/13), and no patients died of ESCC.

In the overall cohort of 104 patients, 8 patients demonstrated LNM or distant metastasis, as summarized in Table 4. Four patients had LNM that was revealed by surgical resection as an additional therapy. Moreover, four patients experienced recurrence, and the median recurrence period was 22 months (range: 10–31) after ER. Only patients with LNM survived with surgical lymph node dissection; however, no patients with distant metastasis survived (Figure 2).

### 3.4. Factors for Metastasis and Survival in pT1a-MM ESCC

The lesion size was the only significant influencer on disease-free survival, as shown in Table 5. Lesion size was compared in those with and without metastatic disease and was found to be significantly greater in metastatic patients vs. those without metastatic disease (41 (25–94) mm vs. 26 (4–75) mm; *p* < 0.05) respectively. All ESCCs which had metastasis were more than 25 mm in size. The Cox proportion hazard model showed that the presence of HNC (hazard ratio: 5.16, 95% confidence interval (CI): 1.64–16.89), chemotherapy for HNC (hazard ratio 5.06, 95% CI: 1.01–24.44), and metastasis (hazard ratio: 45.41, 95% CI: 6.93–304.05) were the factors associated with overall survival (Table 5).

### 3.5. Long-Term Outcomes

The attrition rate during follow-up was 1.9% (2/104 lost to follow-up). Moreover, during the follow-up period, 20 patients perished. ESCC did not result in any mortality in Group 4 patients, in whom mortality was solely attributed to HNC. The five-year overall survival rates in Groups 1, 2, 3, and 4 were 88.9%, 93.8%, 91.7%, and 75.1%, respectively (Figure 3A). The Cox proportional hazard model showed that HNC was a significant predictor of survival (hazard ratio: 3.95, 95% CI: 1.48–12.38). The five-year, disease-specific survival rates in Groups 1, 2, 3, and 4 were 94.2%, 93.8%, 100%, and 100%, respectively (Figure 3B). The Cox proportional hazard model did not show any differences among the four groups (*p* = 0.493).

## 4. Discussion

To the best of our knowledge, this is the first study to report the cumulative metastasis rate of pT1a-MM ESCC based on the IHC evaluation of LVI in a large cohort (>100) patients. Some reports have shown the long-term outcomes of pT1a-MM ESCC; however, the LVI was not evaluated by IHC for all lesions [5,12,13]. The cumulative metastasis rates of pT1a-MM ESCC in all patients and patients without LVI or HNC were 7.7% (8/104) and 7.3% (3/41), respectively. These metastasis rates are similar to those in previous reports in which LVI was evaluated by HE staining [5,8,9,10,12,13]. As previously reported, LVI detection increased after adding IHC [14], and the rate of LVI in this study (43.3%) was higher than that in the previous report (16.6%) [12]. After excluding patients with HNC, we found a more favorable long-term outcome of pT1a-MM ESCC regardless of LVI or additional prophylactic therapy, and no patient died of HNC.

The positive effect of IHC in the evaluation of LVI was limited in this study. Although the addition of IHC to 33 patients identified eight additional LVI cases, the associated rate of metastasis involvement did not increase significantly and only increased from 6.1% (3/49) to 7.3% (3/41) in Group 1. Indeed, all LVI patients identified by IHC did not develop metastatic disease during follow-up, and the three patients with metastatic disease in Group 1 did not show LVI by IHC. While identifying LVI may have incurred additional therapy for these patients, such therapy may have been superfluous, as metastatic disease did not develop. Therefore, the decision to add IHC during the pathological evaluation of LVI remains at the discretion of the pathologist [8]. Based on our data, the prognostication of patients did not change when IHC was added to the previously performed HE staining, as has been reported previously [4,5,9,10,11].

Consistent with previous studies showing that increased tumor size (>15 mm) is an independent factor for metastatic risk [9,10,11], our data showed that ESCC > 25 mm is a significant factor associated with increased metastatic risk (Table 5). This may relate to an increase in the points of contact with the lymphovascular ducts as tumor size increases [23,24]. It should be noted that the evaluation of LVI is inherently limited by the width of the sections (2 mm) made during gross pathology preparation of the lesion. Therefore, it is possible to skip LVI areas during grossing, and the absence of LVI cannot be ascertained, as it may be haphazardly captured during the microtome process. To better risk stratify the development of metastatic disease, other factors may be involved, such as micro-RNA or exosome sequencing, which may help increase the accuracy of metastasis prediction [9,10,11,25,26]. Even though the Japan Gastroenterological Endoscopy Society ESD/EMR guidelines do not recommend for or against the additional treatments in patients with pT1a-MM ESCC without LVI, additional therapy may be considered when ESCC is larger than 25 mm in size regardless of LVI, gleaning the results of this study. A previous study reported that lymphoid invasion but not lesion size was an independent factor for metastasis [13]. However, this study did not evaluate LVI based on IHC results. In our research, the effect of LVI on metastasis might have decreased because there was no advantage of IHC-based LVI for predicting metastasis. The advantage of lesion size might be that it is relatively smaller than that of lymphoid invasion in a previous study.

In our series, additional prophylactic therapies were not associated with improved prognosis, unlike previous reports [5,12], where patients who did not receive additional therapy were also those with worse comorbid conditions. To improve risk stratification in our study, we separately analyzed patients with HNC who were likely to be in worse condition. It is also important to note that previous studies amalgamated patients with pT1a-MM and pT1b-SM1 ESCC and analyzed their outcomes as a single group. Therefore, previous studies may have been confounded by the depth of tumor invasion and performance status. Importantly, we demonstrated that the presence of LVI does not result in worse outcomes in patients with pT1a-MM ESCC, and the worse long-term prognosis of these patients is mainly related to the presence of concomitant HNC rather than LVI.

In our follow-up, recurrence occurred in four patients, three of whom had esophageal cancer, and salvage therapy was only successful in one patient through resection of LNM. Distant metastatic recurrences were detected using CT within three years after ER. Although endoscopic ultrasonography has been purported to be useful in detecting metastasis [27], its utility in distant metastasis detection is limited. On the other hand, while positron emission tomography-CT may help capture distant metastasis earlier, its use is hampered by cost and accessibility. Future research should focus on methods that allow the prediction of metastasis earlier and more accurately, regardless of LVI status [25,26].

Our study had several limitations that should be acknowledged, such as the retrospective study design and the relatively small sample size. In addition, our center did not have a standardized protocol for providing additional therapies, and this was offered after shared decision-making with the patient, tailored uniquely to each patient. We compensated for these potential shortcomings by selecting a homogeneous study group restricted to a well-defined subset of ESCC. Furthermore, other than implementing C-EMR for earlier patients, our ER method and patient care were uniform over the study period, decreasing the likelihood of an era effect. Another limitation may be related to the patients who received a telephone survey rather than inpatient visits. However, the telephone surveys only contacted the families of the two patients who had already died. Furthermore, less than 2% of the patients were lost to follow-up, minimizing observation bias. Finally, we excluded patients with HNC and generated a more homogenous study group for prognostic analysis.

## 5. Conclusions

In this large-scale, single-center study of patients with ESCC, we determined the metastasis rate of pT1a-MM ESCC based on LVI evaluated by IHC and additional therapy. The advantage of IHC-based evaluation for LVI was limited in determining the necessity of additional prophylactic therapy for pT1a-MM ESCC patients, and future studies should evaluate more accurate prognostification tools that permit an earlier and more reliable prediction of distant metastasis in superficial squamous cell esophageal cancer.

## Figures and Tables

**Figure 1 jcm-11-06969-f001:**
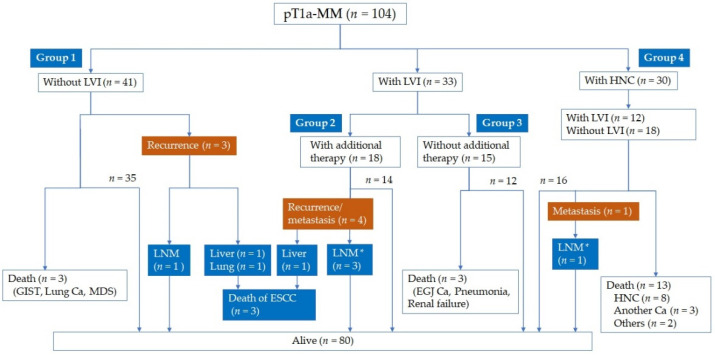
The clinical course of patients. LNM, lymph node metastasis; MDS, myelodysplastic syndrome; EGJ Ca, esophagogastric junctional carcinoma; Ca, carcinoma. * surgical resection as additional prophylactic therapy revealed LNM.

**Figure 2 jcm-11-06969-f002:**
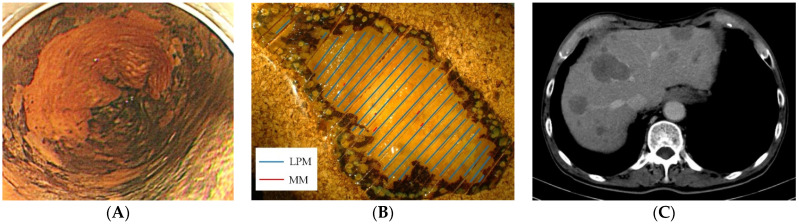
A case of superficial squamous cell carcinoma invading the MM without LVI showed liver metastasis after ER (Case 6). (**A**) The lesion showed a large unstained area under Lugol chromoendoscopy. (**B**) The lesion was endoscopically resected, and the pathology showed squamous cell carcinoma invading up to the MM; LVI was negative by IHC. (**C**) CT 32 months after ER showed multiple liver metastases. The biopsy from the liver showed squamous cell carcinoma. The patient died of liver metastasis 38 months after ER.

**Figure 3 jcm-11-06969-f003:**
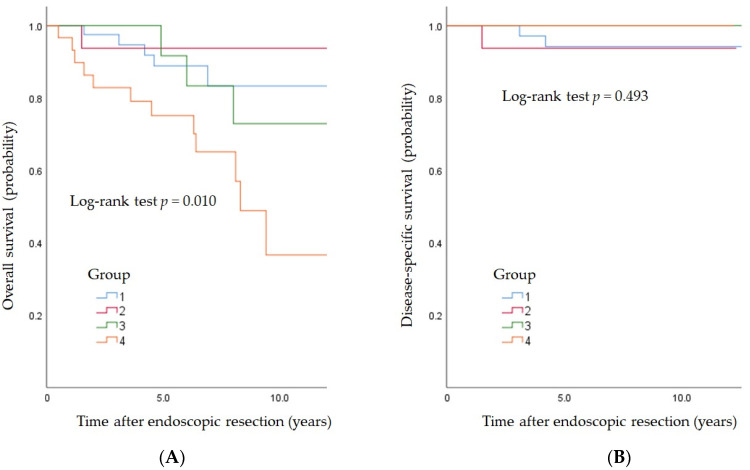
(**A**) The overall survival. The five-year survival rate (SE) in each group was as follows: Group 1, 0.889 (0.053); Group 2, 0.938 (0.061); Group 3, 0.917 (0.080); and Group 4, 0.751 (0.082). The hazard ratio (95% CI) of each group compared to Group 1 was as follows: Group 2, 0.64 (0.03–3.96); Group 3, 1.36 (0.28–5.59); Group 4, 3.95 (1.48–12.38). (**B**) The disease-specific survival. The five-year, disease-specific survival (SE) in each group was as follows: Group 1, 0.942 (0.040); Group 2, 0.938 (0.061); Group 3, 1.000 (0.000); and Group 4, 1.000 (0.000). The hazard ratio (95% CI) of each group compared to Group 1 was as follows: Group 2, 1.48 (0.07–15.53); Group 3, 0.00 (0.00–4.58); Group 4, 0.00 (0.00–2.53).

**Table 1 jcm-11-06969-t001:** The baseline characteristics of patients who received ER for superficial pT1a-MM ESCC.

	Total	Group 1	Group 2	Group 3	Group 4	*p*
Number	104	41	18	15	30	
Sex, male/female	94/10	35/6	17/1	13/2	29/1	0.372
Age, mean ± SD, years	64.8 ± 8.6	65.1 ± 8.9	65.5 ± 9.9	65.7 ± 7.3	63.4 ± 8.1	0.785
Tumor size, mean ± SD, mm	29.7 ± 16.4	27.8 ± 16.4	34.7 ± 12.4	32.3 ± 16.3	28.1 ± 18.5	0.423
Morphological type, IIc	42 (40.4%)	15 (36.6%)	7 (38.9%)	9 (60%)	11 (36.7%)	0.418
Infiltrative growth pattern, c	2 (1.9%)	0	0	0	2 (6.7%)	0.170
Lymphovascular invasion	45 (43.3%)	0	18 (100%)	15 (100%)	12 (40%)	<0.001
Additional prophylactic therapy	24 (23.1%)	0	18 (100%)	0	6 (20%)	<0.001
Distant or lymph node metastasis	8 (7.7%)	3 (7.3%)	4 (22.2%)	0	1 (3.3%)	0.056
Death from all causes	22 (21.2%)	5 (12.2%)	1 (5.6%)	3 (20%)	13 (43.3%)	0.004
Death from esophageal cancer	3 (2.9%)	2 (4.9%)	1 (5.6%)	0	0	0.498

SD, standard deviation.

**Table 2 jcm-11-06969-t002:** The rate of LVI based on HE and IHC staining.

	Only HE (*n* = 33)	HE + IHC (*n* = 33)
Lymphoid invasion	5	10
Vascular invasion	1	8
Lymphoid or vascular invasion	5 (15.1%)	13 (39.4%)

LVI, lymphovascular invasion; HE, hematoxylin and eosin; IHC, immunohistochemical.

**Table 3 jcm-11-06969-t003:** The rate of metastasis based on LVI and additional therapy.

	LVI (−) Additional Therapy (−)	LVI (+) Additional Therapy (+)	LVI (+) Additional Therapy (−)
Rate of metastasis	5.1% (3/59) *	20.8% (5/24)	0% (0/21)
		11.1% (5/45) *	

* No difference was observed in the metastasis rates of pT1a-MM with and without LVI. LVI, lymphovascular invasion.

**Table 4 jcm-11-06969-t004:** Patients with recurrent metastatic disease after ER.

Case	Age	Sex	Location	Size (mm)	Type	ER	ly	v	Additional Therapy	Duration of Recurrence (Months)	Metastasis	Additional Therapy after Surgery or Recurrence	Status
1	73	M	Ut	46	IIb	ESD	0	1	SR	-	LN (#105)	CT	Alive
2	46	M	Mt	36	IIb	ESD	1	0	SR	-	LN (#1)	CT	Alive
3	64	M	Lt	45	IIc	ESD	1	0	SR	-	LN (#2)	None	Alive
4	61	M	Lt	94	IIb	ESD	1	0	SR	-	LN (#9)	CT	Alive
5	60	M	Mt	27	IIb	ESD	1	0	RT	10	Liver LN (#110)	CT	Death of ESCC
6	66	M	Lt	56	IIb	ESD	0	0	-	31	Liver	CT	Death of ESCC
7	63	F	Lt	25	IIa	ESD	0	0	-	28	Lung	CT	Death of ESCC
8	60	M	Mt	33	IIa	ESD	0	0	-	15	LN (#2)	CT + SR	Alive

M, male; F, female; Ut, upper thoracic esophagus; Mt, middle thoracic esophagus; Lt, lower thoracic esophagus; ESD, endoscopic submucosal dissection; SR, surgical resection; RT, radiation therapy; LN, lymph node; CT, chemotherapy; ESCC, esophageal squamous cell carcinoma; ly, lymphovascular invasion; v, vascular invasion.

**Table 5 jcm-11-06969-t005:** Factors associated with disease-free survival and overall survival.

	Disease-Free Survival, Hazard Ratio (95% CI)	Overall Survival, Hazard Ratio (95% CI)
Age	0.98 (0.93–1.03)	1.03 (0.97–1.10)
Tumor size, per 10 mm	1.31 (1.01–1.67)	1.12 (0.81–1.53)
Morphological type, IIc	0.75 (0.29–1.87)	1.04 (0.34–3.21)
Infiltrative growth pattern, c	1.47 (0.19–7.81)	0.92 (0.11–5.67)
Lymphovascular invasion	1.48 (0.50–4.17)	2.77 (0.83–9.39)
Additional prophylactic therapy	1.14(0.37–3.49)	0.50 (0.10–1.96)
Head and neck cancer	2.22 (0.85–5.41)	5.16 (1.64–16.89)
Chemotherapy for head and neck cancer	2.01 (0.46–7.85)	5.06 (1.01–24.44)
Lymph node metastasis or recurrence	-	45.41 (6.93–304.05)

Tumor size was extracted for the independent factor for disease-free survival. The history of HNC, chemotherapy for HNC, and lymph node metastasis or recurrence were selected for independent factors for overall survival. CI: confidence interval.

## Data Availability

The data that support the findings of this study are available from the corresponding author upon reasonable request.

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
