# Peer review of "The Advantage of Immunohistochemical Staining for Evaluating Lymphovascular Invasion Is Limited for Patients with Esophageal Squamous Cell Carcinoma Invading the Muscularis Mucosa"

_jcm, 2022, doi:10.3390/jcm11236969_

Round 1

Reviewer 1 Report

This is a well-organized retrospectively study including 104 patients with endoscopically resected pT1a-MM ESCC. There are some major concerns should be revised in this manuscript.

1.Because the exclusion criteria defined “a history of other neoplasms treated with any form of chemotherapy or radiotherapy before ER”.

The definition of “HNC” in the group 4 of current study should be clearly defined in the M&M, including synchronously or metachronously with ESCC ? Or previous HNC history before ER?

2.Also, please clearly define the “additional prophylactic therapy” in the 2.5 Additional prophylactic therapy.

At least describe the type of surgery, the field of radiotherapy, the regimen of chemotherapy

3. Although advantage of IHC for determining additional prophylactic therapy is limited in the current study. Please further discuss the “indication” of additional prophylactic therapy”, for example, margins, R1 resection, or tumor size in the “Discussion”.

4. Please check the Table 5. The OR (95%CI) of the factor “size” is missing.

5.As the authors acknowledged in the limitation, the current study did not have a standardized protocol for providing additional therapies.

The major drawback of this study comes from the grouping mixed with patients “without or with LVI” and “with additional therapy and without additional therapy“. Selection biases existed in the group 2 which may lead to confounding factors in the subsequent analyses.

In addition, in the analyses of factors for metastases, patients who received chemotherapy or radiotherapy for ESCC (group 2 ?) or HNC (group 4) and those who did not have metastasis (n=35) were excluded

Therefore, a direct comparison of factors between “group 1 and group 3” should be more accurate and appropriate for the purpose of this study.

The authors should “directly” analyzed factors for survival and metastases between “Group 1 and Group 3” using Kaplan–Meier survival and Cox proportional hazard model.

I am glad to review the revised manuscript.

Author Response

We would like to thank the reviewers for their time and comprehensive feedback. Our responses follow.

Reviewer 1

  1. Because the exclusion criteria defined “a history of other neoplasms treated with any form of chemotherapy or radiotherapy before ER”.

The definition of “HNC” in the group 4 of current study should be clearly defined in the M&M, including synchronously or metachronously with ESCC ? Or previous HNC history before ER? Synchronously pharyngeal SCC was no excluded.

Reply: We thank the reviewer for pointing out this issue. We excluded patients who had a past history of HNC but included synchronous HNC which was treated after ER for ESCC. We add this sentence in the section “2.2 Study group.”

2.Also, please clearly define the “additional prophylactic therapy” in the 2.5 Additional prophylactic therapy. At least describe the type of surgery, the field of radiotherapy, the regimen of chemotherapy

Reply: We thank the reviewer for the suggestion. We have specified esophagectomy was performed as the surgical treatment in sentences added to the section “2.5 Additional prophylactic therapy.”

  1. Although advantage of IHC for determining additional prophylactic therapy is limited in the current study. Please further discuss the “indication” of additional prophylactic therapy”, for example, margins, R1 resection, or tumor size in the “Discussion”.

Reply: We thank the reviewer for the suggestion. We added a sentence in the Discussion at lines 375–378.

  1. Please check the Table 5. The OR (95%CI) of the factor “size” is missing.

Reply: Table 5 was replaced in the revised manuscript.

5.As the authors acknowledged in the limitation, the current study did not have a standardized protocol for providing additional therapies. The major drawback of this study comes from the grouping mixed with patients “without or with LVI” and “with additional therapy and without additional therapy“. Selection biases existed in the group 2 which may lead to confounding factors in the subsequent analyses. In addition, in the analyses of factors for metastases, patients who received chemotherapy or radiotherapy for ESCC (group 2 ?) or HNC (group 4) and those who did not have metastasis (n=35) were excluded Therefore, a direct comparison of factors between “group 1 and group 3” should be more accurate and appropriate for the purpose of this study.

The authors should “directly” analyzed factors for survival and metastases between “Group 1 and Group 3” using Kaplan–Meier survival and Cox proportional hazard model.

Reply:

We thank the reviewer for the suggestion. According to the suggestion, we changed the statistical analysis. We eliminated the multivariate analyses and conduct analyses with Cox proportional hazard model. We edited table 5 accordingly.

Reviewer 2 Report

Fist, I want to congratulate the authors for their work. Here are my comments:

1) In Introduction the authors should define the endoscopic resection and the differences between endoscopic mucosal resection and endoscopic submucosal resection.

2) The authors analyzed 104 lesions in 104 patients (one lesion per patient). Are patients with more than one lesion? Are patient with more than one lesion excluded from this study?

Was the lesion taken as one specimen or more specimens?

3)The authors should specifiy the length of follow up.

Author Response

We would like to thank the reviewers for their time and comprehensive feedback. Our responses follow.

Reviewer 2

1) In Introduction the authors should define the endoscopic resection and the differences between endoscopic mucosal resection and endoscopic submucosal resection.

Reply: We thank the reviewer for the suggestion. We added some sentences in Introduction at lines 38–44.

2) The authors analyzed 104 lesions in 104 patients (one lesion per patient). Are patients with more than one lesion? Are patient with more than one lesion excluded from this study? Was the lesion taken as one specimen or more specimens?

Reply: We thank the reviewer for pointing out this issue. We also excluded patients in whom more than two pT1a-MM ESCCs were treated. However, there was no patient who had more than two pT1a-MM ESCCs. We added this sentence to the Materials and methods section “2.2 Study group.”

3)The authors should specify the length of follow up.

Reply: We described the last of follow-up in the Materials and methods section “2.7 Statistical analysis” as follows: The last follow-up was defined as either the date of death or the date of last contact obtained from medical records (in-person visit) or a telephone interview. We also mention the median and range of follow up in the results section on line 175.